

# A lightweight intrusion detection method for IoT based on deep learning and dynamic quantization

Zhendong Wang[1], Hui Chen[1], Shuxin Yang[1], Xiao Luo[2], Dahai Li[1] and Junling Wang[1]

[1] School of Information Engineering, Jiangxi University of Science and Technology, Ganzhou, China
[2] School of Electrical Engineering ang Automation, Jiangxi University of Science and Technology, Ganzhou, China

## ABSTRACT

Intrusion detection ensures that IoT can protect itself against malicious intrusions in extensive and intricate network traffic data. In recent years, deep learning has been extensively and effectively employed in IoT intrusion detection. However, the limited computing power and storage space of IoT devices restrict the feasibility of deploying resource-intensive intrusion detection systems on them. This article introduces the DL-BiLSTM lightweight IoT intrusion detection model. By combining deep neural networks (DNNs) and bidirectional long short-term memory networks (BiLSTMs), the model enables nonlinear and bidirectional long-distance feature extraction of complex network information. This capability allows the system to capture complex patterns and behaviors related to cyber-attacks, thus enhancing detection performance. To address the resource constraints of IoT devices, the model utilizes the incremental principal component analysis (IPCA) algorithm for feature dimensionality reduction. Additionally, dynamic quantization is employed to trim the specified cell structure of the model, thereby reducing the computational burden on IoT devices while preserving accurate detection capability. The experimental results on the benchmark datasets CIC IDS2017, N-BaIoT, and CICIoT2023 demonstrate that DL-BiLSTM surpasses traditional deep learning models and cutting-edge detection techniques in terms of detection performance, while maintaining a lower model complexity.

# INTRODUCTION

The technology of the Internet of Things (IoT) is advancing rapidly in the current century. Due to its ability to allow autonomous data transmission without the need for human intervention, it has a wide variety of applications in many industries and fields, including agriculture, unmanned vehicles, healthcare, and industry. The IoT is anticipated to revolutionize the way humans live in the future, aiming to create more autonomous and fully intelligent systems (*Nguyen et al., 2021*; *Castiglione et al., 2021*; *Chen & Yang, 2019*; *Khan, Silva & Han, 2016*; *Gope & Sikdar, 2018*). Nevertheless, the publicly deployed

Corresponding author
Hui Chen,
6120210177@mail.jxust.edu.cn

nature of the existing IoT device environment renders them vulnerable to physical and cyber attacks. Security issues, such as user privacy leakage caused by cyber attacks, can be particularly detrimental. Therefore, it is urgent to establish a robust and adaptive intrusion detection system to address the security issues in the IoT.

Intrusion detection systems are composed of three primary components: data collection, conversion to select features, and decision engines. Data collection involves obtaining different types of network data from the network environment, such as system calls or network traffic. The quality and quantity of the data are crucial for accuracy and efficiency, as they can establish a baseline for normal network behavior and detect any abnormal or suspicious intrusions in current activities. Conversion to select features is the process of representing predefined units of data, such as system calls in processes or data streams within time windows, as attribute lists, also known as feature vectors. This step involves extracting relevant information, reducing dimensions, and selecting features with maximum information content to achieve effective intrusion detection. It helps represent the data in a way that determines discriminative attributes while reducing complexity and improving detection performance. The decision engine is an algorithm or heuristic method used to determine whether given data should be considered an attack. Depending on the type of decision engine used, intrusion detection can be categorized into misuse, anomaly, or hybrid (*Bridges et al., 2019*).

Misuse intrusion detection filters events by defining attack characteristics. However, emerging unknown attacks can easily escape the markers of misuse intrusion detectors, making it necessary to periodically update the types of attacks in the database used by the detectors. Anomalous intrusion detection discriminates potential attacks based on normal behavior learned over time. Compared to misuse intrusion detection, anomalous intrusion detection can detect attacks that have not appeared before. However, anomalous intrusion detection is prone to false alarms, which can lead to a reduction in detection accuracy. Second, if attacks are included in the training data, they can easily bypass the detector. Hybrid intrusion detection is a technique that aims to overcome the limitations of a single detection method by combining the first two detection methods. The hybrid detector is trained to classify known attacks and normal behavior and can identify both known and unknown attacks during the detection process. By adopting this approach, the detection accuracy can be significantly improved as it reduces the number of false positives generated by anomaly detection. Hybrid intrusion detection combines the strengths of two detection methods to create a more comprehensive and robust defense mechanism against attacks on IoT systems.

Due to the diversity of network attacks and the complexity of the attack environment, deep learning has become an effective technique to improve the performance of network intrusion detection (*Lansky et al., 2021*; *Tharewal et al., 2022*; *Wang et al., 2021*; *Khan et al., 2021*; *Mahbooba et al., 2021*). While deep learning can better capture complex nonlinear features in network traffic data due to its superior performance, the deployment of IoT devices must take into account various cost, scale, and power constraints (*Albulayhi et al., 2021*; *Rehman et al., 2022*; *Nguyen et al., 2022*). Furthermore, most deep learning models are computationally complex, limiting their ability to function effectively on IoT devices

with limited computational resources and potentially causing damage to the devices. Therefore, it's important to pay close attention to the following goals when developing an IoT intrusion detection system (*Sharma & Verma, 2021*; *Roy et al., 2022*).

1. Flexible data processing: By employing optimization methods, such as data multicollinearity elimination, sampling, feature extraction, and dimensionality reduction, the intrusion detection system can identify network attacks in a shorter training time.

2. Small model and low energy consumption: The optimization and compression of the model minimize the energy consumption and computational resource loss of the intrusion detection system, enabling its deployment in network entities with low computational and storage capacity. This approach is more suitable for the actual situation where IoT devices have limited resources.

3. Strong detection performance: The security performance of IoT devices will significantly impact the user's trust in the device. To minimize the risk of security attacks in a network and protect users from potential property and reputation loss, it is essential to develop an intrusion detection system with robust detection capabilities.

To accomplish the goals outlined above, a lightweight neural network model combining high detection performance and low model complexity is suggested in this article. In previous research, bidirectional long short-term memory (BiLSTM) networks have been widely used for sequence feature correlation learning and have performed well in intrusion detection. However, it tends to focus more on feature extraction of bi-directional long-distance dependent information and neglects the mining of deep information features, which reduces the sensitivity of detection of attacks with complex nonlinear features. Therefore, in this article, deep neural networks (DNNs) are integrated into BiLSTM to enhance its ability to extract nonlinear features from deep information. Although this fusion can improve the detection performance of the model, it will also increase the number of model parameters and complexity, which contradicts the aforementioned goals. To solve this problem, this study further uses dynamic quantization methods to compress specified network structure units to generate a more streamlined intrusion detection model. We also use the incremental principal component analysis (IPCA) algorithm to eliminate noise and redundant information in the high-dimensional space of the original data so as to filter out high-quality feature subsets. To make the model reach its optimal state during the training process, the optuna optimization algorithm is introduced to adjust hyperparameters such as neural network loop weights, training input batch size, learning rate to to improve the model detection performance. The proposed model is subjected to relevant experiments on the benchmark datasets CIC IDS2017, N-BaIoT, and CICIoT2023, and the experimental results demonstrate that the proposed model exhibits excellent detection performance while incurring less computational resource loss than traditional deep learning models.

Overall,the proposed work mainly makes the following four contributions:

- An IoT intrusion detection model that combines deep neural network and bidirectional long short-term memory neural network (DNN-BiLSTM) is designed. This model uses

DNN to perform nonlinear feature extraction and BiLSTM to extract long-distance dependent information in both directions, resulting in more effective feature extraction.

- A lightweight IoT intrusion detection method based on neural network model quantization is proposed. This method applies dynamic quantization to DNN-BiLSTM models, which addresses the problem of increasing model complexity caused by enhanced feature extraction capability without significantly affecting detection performance.
- The incremental principal component analysis (IPCA) algorithm is introduced to achieve feature dimensionality reduction. IPCA can incrementally learn data features and is more suitable for the characteristics of real-time monitoring of network traffic in the IoT environment.
- An optuna framework based on Bayesian optimization is designed to optimize model hyperparameters, facilitating the determination of gradient boosting schemes and maximizing neural network performance.

The remaining portions of this essay are organized as follows: A thorough survey of the literature on prior IoT intrusion detection studies is provided in the Literature review section. The proposed lightweight IoT intrusion detection method, which includes the detection model, feature reduction algorithm, model quantization technique, and model hyperparameter optimization framework, is comprehensively discussed in the Method and methodology section. In the Datasets and Preprocessing section, we present the dataset and preprocessing methods employed in the experiments. The Experimental results and analysis section investigates the proposed method's performance in IoT detection and intrusion detection's lightweight capabilities. Finally, in the concluding section, we present the research's conclusions and suggest some future research directions.

## LITERATURE REVIEW

This part will concentrate on the current intrusion detection models used in the IoT and incorporate fresh designs and developments of lightweight IoT intrusion detection schemes.

### Intrusion detection solutions for the IoT

With the widespread adoption of the IoT in various fields, many scientific techniques have been deployed by researchers for anomaly detection in the IoT, among which machine learning and deep learning techniques are most favored. Initially, the research objectives of IoT intrusion detection focused on model efficiency and reliability; however, most of the system design choices limit the extent to which this goal can be achieved, such as pursuing reliability at the expense of dataset selection and feature processing or pursuing only efficiency at the expense of model overfitting problems. Therefore, *Derhab et al. (2020)* advocated five design principles for IoT intrusion detection solutions and designed and implemented a deep learning framework for IoT intrusion detection systems, temporal convolutional neural network (TCNN), which combines CNN with causal convolution to solve the problem of unbalanced datasets and feature space reduction and feature transformation to achieve efficient feature engineering requirements. TCNN performs well

when compared to some popular machine learning and deep learning approaches when tested on the Bot-IoT dataset.

Wireless communication is an essential IoT building block, but despite the technology's maturity, IoT security concerns related to wireless communication are challenging because of the increasingly complex channel environment. The UNSW-NB15 dataset was chosen by *Tian, Li & Liu (2019)* as a research object. They combined deep learning techniques with shallow learning techniques, using deep autoencoder techniques to reduce the data's dimensionality, and then fed the compressed data into a support vector machine (SVM) to find the optimal SVM classifier parameters with the aid of an artificial bee colony (ABC) algorithm, confirming that the suggested scheme is superior to the PCA-based approach and other methods based on machine learning. However, as more time went by during training, this approach also progressively revealed its flaws.

Cloud computing is a part of storing IoT data, but the different technologies and protocols used lead to complex security issues in the transition from IoT to cloud platforms. *Fatani et al. (2021)* developed a new deep learning framework that consists of two parts: the first part uses CNN model for feature extraction of the input dataset, and the second part uses differential evolution (DE) algorithm to enhance the performance of transient search optimization (TSO) to form a new variant TSODE, which is used for feature selection in the second phase of feature engineering. Considering the more heterogeneous nature of the IoT compared to other traditional networks, *Alaiz-Moreton et al. (2019)* used the data storage framework under attack in the mqtt protocol as input and selected nadam, which has the best optimization loss effect, as a faster loss function optimizer after comparing three static gradient descent methods, rmsprop, adam, and nadam. Then, XGBoost, LSTM, and GRU classification models were tested, and all achieved high detection accuracy but only had better classification results on the binary classification problem.

Past studies have shown that neither independent detection schemes nor centralized cloud computing can perfectly relieve the security threats suffered by the IoT, and the emergence of a new distributed intelligence technique, fog computing, has eased the use of IoT computing resources. *Gavel, Raghuvanshi & Tiwari (2021)* proposed a two-axis dimensionality reduction technique based on Kalman filtering and the salp swarm algorithm with a kernel-based extreme learning machine as a multiclass classifier for intrusion detection in IoT networks, which has high detection accuracy on both NSL-KDD and CICIDS2017 datasets. *Jeyanthi & Indrani (2023)* proposed the ACAAS algorithm, which is used to extract important features from the IoTID20 dataset and utilize a recurrent neural network with long short-term memory for attack identification. *Harris et al. (2020)* emphasized the importance of optimal feature selection techniques to improve the classification performance of the algorithm and combined the particle swarm optimization (PSO) search algorithm with other search algorithms for feature selection and validated the results with the J48 classification algorithm, demonstrating that the PSO algorithm and the combination with other search algorithms improved the TPR and accuracy of J48 classification.

## Lightweight Internet of Things intrusion detection model

IoT traffic typically entails significant data volume and memory space requirements. However, resource constraints in specific scenarios pose challenges for implementing intrusion detection systems based on high-performance deep learning algorithms directly on IoT devices. *Hosseininoorbin et al. (2023)* conducted research to explore the feasibility of leveraging Google Edge TPUs for implementing an IoT edge computing intrusion detection system, considering the computational and energy constraints of IoT edge devices. The results demonstrate the potential of TPUs to significantly enhance the system's computational power, enabling the implementation of deep learning algorithms on edge devices. *Popoola et al. (2020)* designed a long short-term memory autoencoder approach to generate low-dimensional latent spatial feature representations for complex high-dimensional data in the hidden layer. BiLSTM was used for network traffic classification on the classical botnet dataset BoT-IoT for binary and multiclassification experiments, demonstrating the highest data reduction rate of 91.89% for this hybrid approach. *Murali & Jamalipour (2019)* proposed a novel ABC-based lightweight intrusion detection algorithm for analyzing Sybil attacks in mobile low-power and lossy networks from a bioinspired perspective. The study proved that this lightweight intrusion detection algorithm has accuracy, sensitivity and effectiveness in all three types of sybil attacks.

*Sudqi Khater et al. (2019)* divided lightweight research work into three stages: data preprocessing, model selection, and performance evaluation. Feature extraction is done using an improved vector space representation technique called n-gram method, and mutual information feature selection is used to reduce the number of features. *Soe et al. (2020)* proposed the gain ratio correlation set thresholding algorithm as a feature selection algorithm to make the intrusion detection system lightweight. The feature selection algorithm was used for several tree-based classifiers for experiments on the BoT-IoT dataset and verified that the J48 algorithm is the most suitable for the IoT environment in terms of lightweight and detection performance. *Davahli, Shamsi & Abaei (2020)* proposed a hybrid genetic algorithm (GA) and grey wolf optimization algorithm (GWO) dimensionality reduction algorithm, which was applied to the real wireless dataset AWID for experiments to define the most pertinent wireless traffic features with support vector machines as classifiers. The results confirmed that the GA-GWO algorithm diminished the dataset's feature dimensionality from 154 to 92 and decreased the computation time. *Lahasan & Samma (2022)* used an efficient two-layer encoder model to synchronously realize the selection of dataset features, training instances, and self-encoder neurons, which combines the advantages of high precision of the K-nearest neighbor classifier and low complexity of the self-encoder. The experimental findings on the IoT intrusion detection dataset N-BaIoT demonstrate that this lightweight model has high anomaly detection accuracy and outperforms most classical optimizers.

From the above several lightweight IoT intrusion detection scheme approaches, it can be summarized that most of the studies have been conducted from feature dimension reduction for preprocessing of datasets and lightweighting of models to reduce the loss of computational resources and memory space for IoT devices. In addition, many researchers have started to introduce the use of optimization algorithms to compensate

for the performance shortcomings caused by model lightweighting. This article suggests an intrusion detection technique that is superior to existing intrusion detection schemes. It uses dynamic quantization and fused DNN-BiLSTM models, and it is applied to the CIC IDS2017, N-BaIoT, and CICIoT2023 datasets. Experiments show that the model design completely takes into account the constraints of limited computational resources and memory space, and it suggests a lightweight and efficient system for identifying physical network attacks.

## MATERIALS AND METHODOLOGY

This section outlines the proposed lightweight IoT intrusion detection technique based on the fusion of DNN and BiLSTM models. First, the general framework of the proposed method is introduced, which includes a comprehensive overview of the proposed lightweight IoT intrusion detection system. Then, the knowledge background of the related techniques is discussed, such as feature reduction algorithms, model quantization methods, the DNN and BiLSTM models used for model fusion, and the Optuna optimization framework. Each technique is described in detail to provide a clear understanding of the proposed lightweight network intrusion detection scheme.

### DL-BiLSTM's overall framework for the IoT intrusion detection method

Numerous current methods for intrusion detection usually rely on reduced machine learning techniques because of the resource constraints associated with the deployment of intrusion detection systems on IoT devices. However, there is an upper bound on the detection accuracy, and missed detections could cause substantial property damage. In this study, a deep learning-based neural network-based intrusion detection scheme is proposed that employs feature dimension reduction and model quantization to simplify the detection model's complexity. The outcome is that the proposed model maintains low computational complexity while achieving high detection accuracy.

Combined with the discussions on modeling techniques in Sections 3.2–3.5, the phases of the proposed approach for IoT intrusion detection in this article are as follows: first, the used IoT intrusion detection dataset is preprocessed, including numericalization of data, data normalization and dimensionality reduction of data using the IPCA algorithm, details of which can be found in Section 4.2. The preprocessed data is then divided into three parts: training set, validation set, and test set. The training and validation sets are used to train the model and fine-tune the weights, and the test set is used to verify the final model classification performance. Next, the training and validation sets are input to DNN-BiLSTM for pretraining, and the model hyperparameters are optimized by the optuna optimization framework to establish the appropriate model parameters and network structure. Finally, the trained optimal network model is quantized by specifying the structural units, the quantized DL-BiLSTM model is output, and the final model detection performance is judged by testing the compressed quantized model using the test set.

The general framework of the proposed lightweight IoT intrusion detection method is shown in Fig. 1, which mainly consists of four modules: data preprocessing, model training and weight fine-tuning, model quantization, and model classification testing.

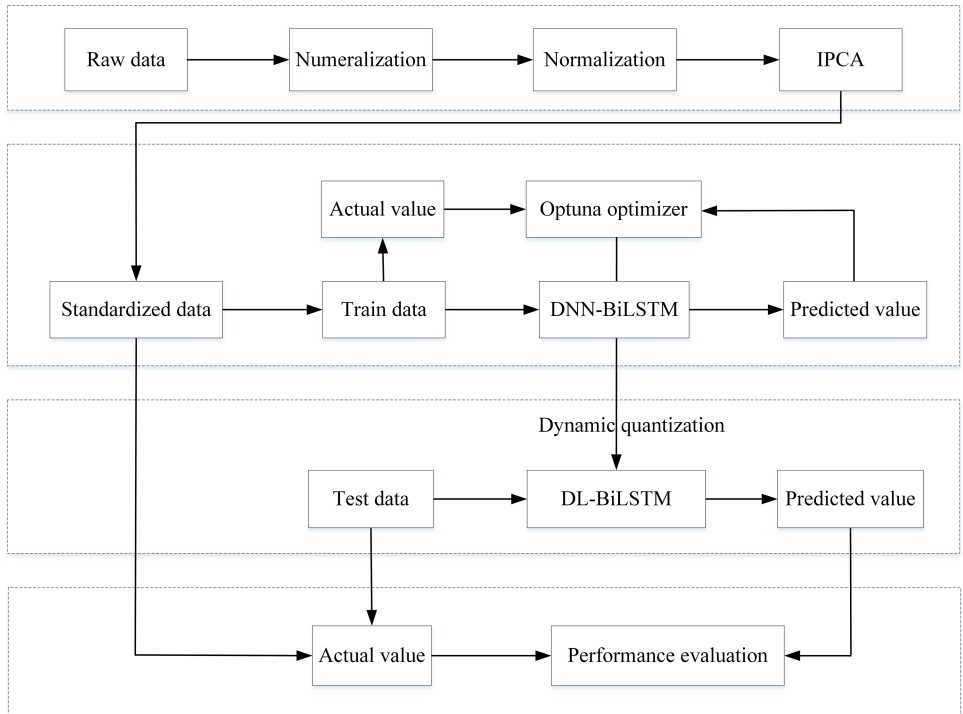

**Figure 1** General framework of lightweight IoT intrusion detection methods.

## IPCA algorithm

When raw traffic data is directly fed into the neural network, irrelevant or redundant features in the raw traffic data typically complicate the feature learning process, which ultimately has a negative impact on the performance of the intrusion detection system as a whole. It is possible that a significant number of instances are needed to fulfill the relationship determination between the input and the target even though the input data already includes enough information because the input space's dimensionality is too high. Both of these problems can be solved by feature reduction, which can be further divided into feature selection, which selects relevant inputs from the original data, and feature extraction, which extracts new variables from the original data that contain the most relevant information to the target variable (*Kwak, Kim & Kim, 2008*).

The most adopted feature reduction method in previous research on intrusion detection systems is the principal component analysis (PCA) algorithm, which ultimately achieves better lightweight performance through this low-complexity linear transformation technique (*Zhao et al., 2021*). To date, the application of some variants of PCA to feature dimensionality reduction of intrusion detection datasets remains hot (*Geetha & Deepa, 2022*). The majority of variations, however, still share the issue that conventional PCA algorithms use batch processing techniques, which call for simultaneous collection of all observed data for computation and have a poor computational efficiency. Especially in the IoT environment where real-time monitoring is required, the complete training data cannot be obtained at one time, and the incremental changes in network traffic can

cause the network attack behavior to change slightly from the original data, resulting in unconvincing data features extracted from the initially given training samples. In this study, a new incremental principal component analysis (IPCA) algorithm for intrusion detection is introduced to replace the PCA algorithm, which is a feature dimensionality reduction algorithm for incremental learning proposed by Hall and Martin to incrementally update feature vectors and feature values (*Hall, Marshall & Martin, 1998*). A brief description of the dimensionality reduction principle of the IPCA algorithm is given below (*Ozawa, Pang & Kasabov, 2008*).

The original high-dimensional data samples are represented as samples $X = (x_1, x_2, \ldots, x_m) \in R^{n \times m}$ in an $n$-dimensional space, and the mean vector $\bar{X}$ of $X$ is calculated from this. $U_k$ is a $n \times k$ dimensional matrix, and $k$ denotes the dimension of the feature space, where the matrix columns correspond to the feature vectors of the data. $\Lambda_k$ denotes a $k \times k$ dimensional matrix with the elements on the diagonal corresponding to the eigenvalues of the data. $\bar{X}$, $U_k$, and $\Lambda_k$ form the eigenspace model $\Omega = (\bar{X})$, $U_k$, $\Lambda_k$, $(n)$ of the original dataset. Suppose a new data sample $Z \in R^n$ is to be added to the original data sample; then, $\bar{X}$ in the original eigenspace model will be updated to $\bar{X}'$, where

$$\bar{X}' = \frac{1}{N+1}\left(N\bar{X} + Z\right) \in R^n. \tag{1}$$

The most critical thing is whether the feature vectors and feature values in the feature vector space need to be updated, and the decision is based on whether the newly added data samples contain energy that does not exist in the previous feature space, based on the following:

$$g = U_k^T (Z - \bar{x}) \tag{2}$$

$$h = (Z - \bar{x}) - U_k g \tag{3}$$

$$\hat{h} = \begin{cases} h/\|h\|, & \text{if } \|h\| > \eta \\ 0, & \text{otherwise} \end{cases} \tag{4}$$

where $h \in R^n$ is the residual vector, and the dimension of the feature space k is added to one when its norm $\|h\|$ is greater than the threshold $\eta$. If the norm $\|h\|$ does not exceed the threshold $\eta$, then the dimension of the feature space maintains k unchanged. If the dimension of the feature space increases, then $U_k$ and $\Lambda_k$ will be changed to $U_{k+1}'$ and $\Lambda_{k+1}'$, respectively, by the following equations:

$$U_{k+1}' = \begin{bmatrix} U_k, & \hat{h} \end{bmatrix} R \tag{5}$$

$$\left\{ \frac{n}{n+1}\begin{bmatrix} \Lambda_k & 0 \\ 0^T & 0 \end{bmatrix} + \frac{n}{(n+1)^2}\begin{bmatrix} gg^T & \gamma g \\ \gamma g^T & \gamma^2 \end{bmatrix} \right\} R = R\Lambda_{k+1}' \tag{6}$$

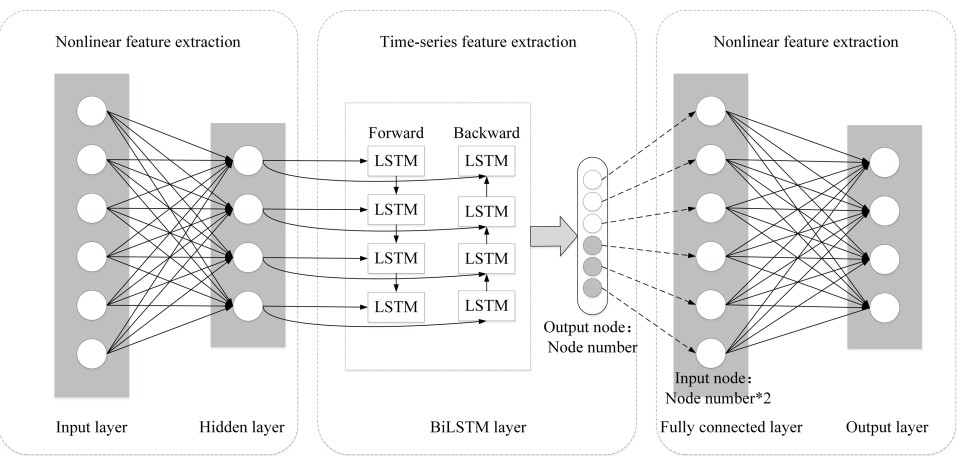

**Figure 2** **Structure of the DNN-BiLSTM model.**

Conversely, if the feature space dimension is not increased, then $U_k$ and $\Lambda_k$ will be updated to $U_k'$ and $\Lambda_k'$:

$$U_k' = U_k R \tag{7}$$

$$\left\{ \frac{n}{n+1} \Lambda_k + \frac{n}{(n+1)^2} g g^T \right\} R = R \Lambda_k'. \tag{8}$$

## DNN-BiLSTM model

In order to improve the ability of BiLSTM to extract nonlinear features and retain its original bidirectional long-distance dependent information feature extraction characteristics, this study designs an intrusion detection model that integrates DNN-BiLSTM. The model structure is presented in Fig. 2.

### *DNN-based nonlinear feature extraction*

Deep neural networks (DNNs) can extract nonlinear features of network information through multiple nonlinear transformations (*Alotaibi et al., 2022*). By applying them to the output of hidden layer neurons, activation functions are usually used to perform nonlinear transformations, enabling the network to learn more complicated functions. The sigmoid, tanh, and ReLU activation functions are frequently employed in neural networks. The sigmoid function maps any real number to a value between 0 and 1, making it suitable for binary classification tasks. The ReLU function returns the same value when the input is positive and 0 when the input is negative, which allows for fast computation and is commonly used in deep neural networks. The tanh function maps any real number to a value between -1 and 1, making it useful in multiclassification tasks. These activation functions extract nonlinear features by mapping the input to a nonlinear space. DNNs enable the network to learn more complex features and functions by nesting multiple nonlinear transformations together.

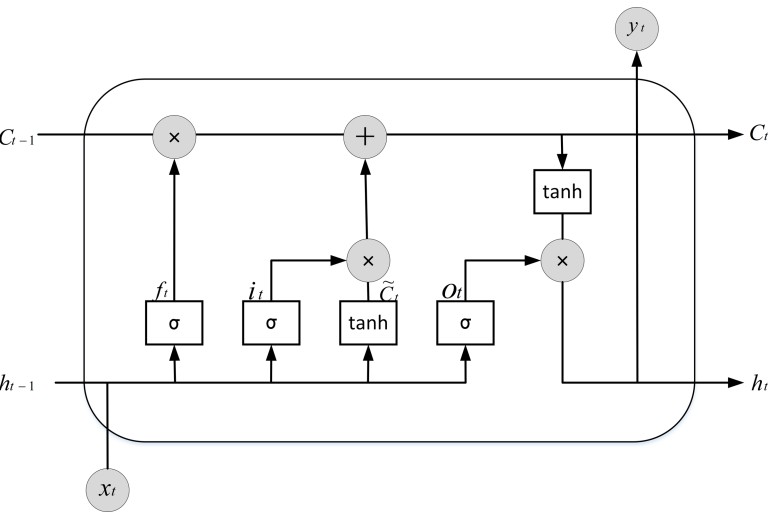

**Figure 3  Internal structure of the LSTM cell.**

### BiLSTM-based temporal feature extraction

The long short-term memory network (LSTM) belongs to the recurrent neural network (RNN) branch. The gradient disappearance issue in RNN brought on by long-term dependencies is resolved by the two main structures that LSTM suggests for storing cells and cell states. These structures allow LSTM to autonomously choose between remembering and forgetting information (*Fu et al., 2022*). The internal structure of the LSTM cell is shown in Fig. 3.

The computational units involved in the process of one time step update of the LSTM occurring are presented below in the form of Eqs.

$$i_t = \sigma(W_{xi}x_t + W_{hi}h_{t-1} + b_i) \tag{9}$$

$$f_t = \sigma(W_{xf}x_t + W_{hf}h_{t-1} + b_f) \tag{10}$$

$$o_t = \sigma(W_{xo}x_t + W_{ho}h_{t-1} + b_o) \tag{11}$$

$$\tilde{c}_t = tanh(W_{xc}x_t + W_{hc}h_{t-1} + b_c) \tag{12}$$

$$c_t = f_t c_{t-1} + i_t \tilde{c}_t \tag{13}$$

$$h_t = o_t tanh(c_t) \tag{14}$$

where $i_t$ denotes the input gate, the current input is $x_t$, and the output of the last hidden layer is $h_{t-1}$, which indicates the hidden state of the previous cell. $\sigma$ denotes the sigmoid activation function, which transforms any value between [0,1]. $f_t$ denotes the forgetting gate, whether the last-minute learned information $c_{t-1}$ can be passed by having it to decide, and $o_t$ denotes the output gate. $\tilde{c}_t$ indicates the candidate memory cell, $c_t$ indicates the current cell state, and $c_{t-1}$ indicates the previous cell state. $h_t$ indicates the hidden state of the current cell. The tanh function is a hyperbolic tangent activation function that converts any value to between $-1$ and 1. The incoming signals pass through the abovementioned gate cells in turn so that the information of the current time step is secured. However, existing studies show that the data features are passed directly from the input to the output of the LSTM, which causes a continuous accumulation of errors and leads to increasing errors as the time step increases.

To solve the defect of error accumulation in the LSTM structure, the neural network structure of bidirectional LSTM is chosen in this article. The bidirectional LSTM consists of two parallel LSTM layers that compute the hidden vectors in opposite directions, and the final prediction results are jointly determined by the forward and backward inputs. This iterative learning of input data features increases the dependency between the data and enhances the memory of sequential information, thereby improving detection accuracy (*Cai et al., 2021*). The feature extraction process of bidirectional BiLSTM can be expressed as Eqs.

$$\overrightarrow{h_t} = tanh(W_{h_t^\rightarrow}x_t + W_{h^\rightarrow h^\rightarrow}\overrightarrow{h_{t-1}} + b_{h^\rightarrow}) \tag{15}$$

$$\overleftarrow{h_t} = tanh(W_{h_t^\leftarrow}x_t + W_{h^\leftarrow h^\leftarrow}\overleftarrow{h_{t-1}}) + b_{h^\leftarrow} \tag{16}$$

$$h_t = \overrightarrow{h_t} + \overleftarrow{h_t} \tag{17}$$

where $h^\rightarrow$ and $h^\leftarrow$ denote the output results of the two forward and reverse LSTMs, $h_t$ denotes the hidden state of the current cell, and the output of the latter hidden layer is $h_{t-1}$, which denotes the output state of the hidden layer of the previous cell. The model structure of BiLSTM is shown in Fig. 4.

## Lightweight IoT intrusion detection model
### Model quantization

Model quantization is an optimization technique that aims to reduce the data bit width of the model parameters from the typical 32-bit floating-point to finite resolution. This technique uses a low bit-width algorithm instead of the full-precision operations commonly used in the past. By compressing the model size, model quantization reduces the consumption of computational resources and improves the inference speed of the model. Typically, model quantization selects a low bit-width of 8 bits to represent the model weights and activations. Previous studies have shown that 8-bit integer quantization can reduce the model size to a quarter of the original size and improve the computational

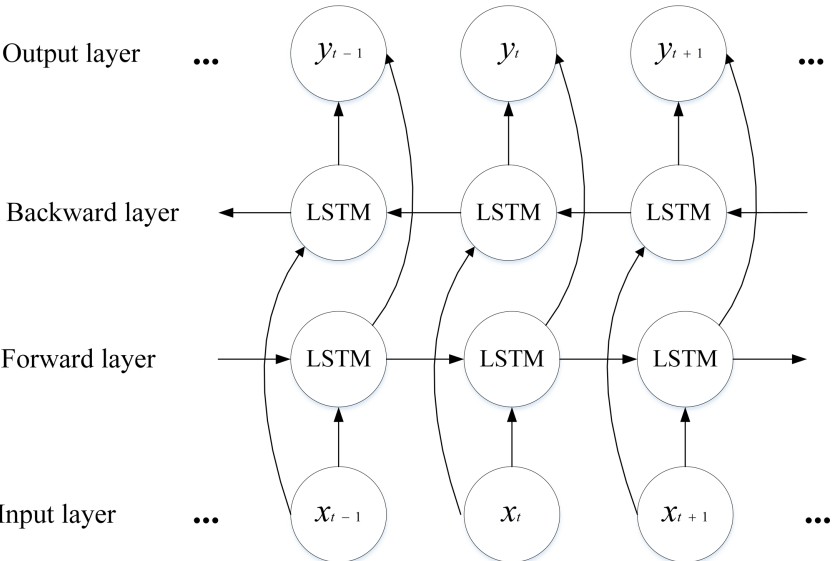

Output layer

Backward layer

Forward layer

Input layer

**Figure 4** **Structure of the BiLSTM model.**

efficiency up to 2-4 times on hardware. Using a bit width lower than 8 bits can further improve computational efficiency, but it comes at a significant cost to model accuracy (*Li, Qu & Wang, 2021*). Hence, we have opted to use an 8-bit integer data bit-width for the quantification of our intrusion detection model.

Various quantization schemes have been proposed in existing studies. Retraining-based quantization is widely adopted and provides efficient hardware implementation and less memory consumption for running neural networks on resource-limited devices, but it requires retraining the neural network model to maintain the detection accuracy of the model (*Kim et al., 2020*). However, retraining the neural network model requires additional computational cost and execution time, which would defeat the purpose of designing a lightweight intrusion detection model. Moreover, frequent access to training data is not feasible in practical scenarios, especially for the IoT environment applied to human life, due to the need to protect user privacy. Therefore, we have chosen posttraining dynamic quantization as our model quantization scheme. Posttraining dynamic quantization involves converting weights to int8, as performed by all quantization types, and requires dynamic conversion of activations to int8, as well as efficient quantization using matrix multiplication and convolution in the computation process. Dynamic quantization can quantize RNNs, LSTMs, and gated cell structures, which is not possible with static quantization. However, dynamic quantization has the drawback of not being able to quantize convolutional and hidden layers. Posttraining dynamic quantization is a more suitable solution for the research setting of this article, which is completely detached from the training process of the dataset and can compress the model size while keeping the accuracy degradation within acceptable limits (*See et al., 2021*). The quantization process

is expressed by the equation.

$$q_{int} = round(float \div scale + offset).$$ (18)

This formula represents the conversion of a floating point value to an integer value by dividing the floating point value by the scale and adding an offset to each level.

$$scale = (float_{max} - float_{min}) \div (q_{max} - q_{min}).$$ (19)

The formula is based on the maximum and minimum values of the original floating-point parameter distribution to obtain the value of. $q_{max}$ and $q_{min}$ denote the maximum and minimum values of the required precision, respectively, and we use 8-bit wide quantization, corresponding to $q_{max} = 127$ and $q_{min} = -128$.

$$offset = q_{min} - (float_{min} \div scale)$$ (20)

This formula is the formula for the offset, which indicates that the floating point has a value of zero relative to the target quantization range before quantization.

### DL-BiLSTM model

This research puts forward a lightweight IoT intrusion detection system based on an intrusion detection model that fuses dual hidden layer DNN with BiLSTM. DNN is able to learn the potential information hidden by features compared with shallow learning models such as machine learning and has a more concise network structure to extract nonlinear features of network data compared with deep learning models. Therefore, DNN is selected as the network structure to optimize the deep nonlinear feature extraction ability of BiLSTM. However, the integration of such multiple network structures tends to increase the number of parameters in the original model, thus consuming more computing resources. As a lightweight IoT intrusion detection system, the size of the detection model and the real-time performance of the detection need to be considered while pursuing higher detection performance.

In this article, we use a dynamic quantization approach to compress the model structure of the fused DNN-BiLSTM model, which reduces the model size and complexity while ensuring the model feature extraction capability and detection classification performance, making the model easier to deploy in resource-constrained IoT environments. This is done by saving and loading the model parameters after the training is completed and subsequently adding a quantizer to perform quantization operations on the trained model. Considering that the current dynamic quantization operation can only effectively quantize structures such as LSTM and Linear, the quantization effect on the Linear structure of the model in this article is quite weak in practice and even brings a high accuracy loss, so the model in this article mainly quantizes the LSTM units in the BiLSTM structure and loads the quantized model and the model parameters saved at the completion of training before the beginning of the testing phase for the evaluation of the model performance in the testing phase. The model structure and generation process of DL-BiLSTM are shown Fig. 5.

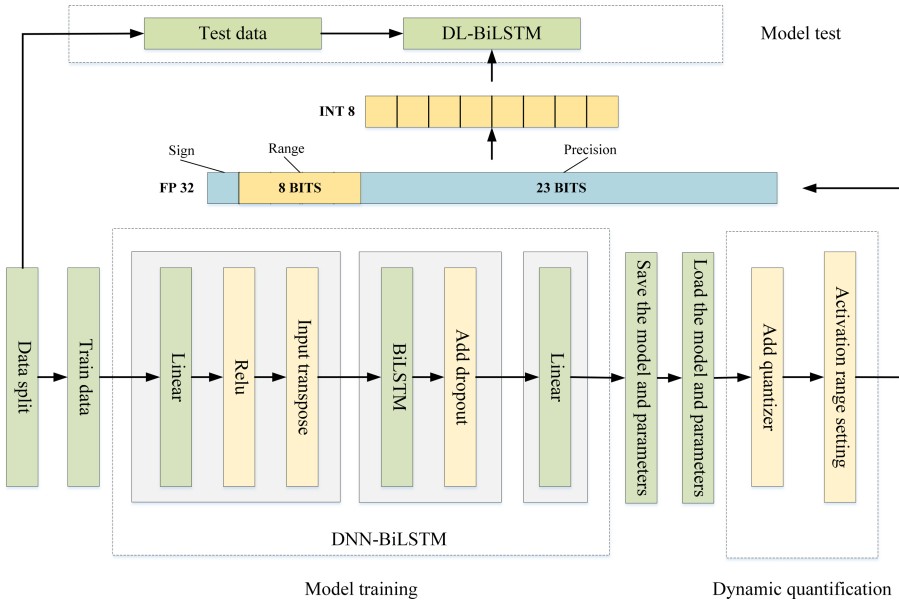

**Figure 5  Structure of the DL-BiLSTM model.**

## Optuna hyperparameter optimization

In this article, the Optuna hyperparameter tuning technique is used to tune hyperparameters related to model detection for better model classification. Optuna is a new generation of hyperparameter optimization frameworks proposed by *Akiba et al. (2019)*. It embodies hyperparameter optimization as a process to find the maximum/minimum value of the objective function, and the objective function returns a verification score for each set of input hyperparameters. Optuna's improvement over traditional manual and automatic tuning techniques is mainly reflected in the following three aspects (*Hanifi et al., 2022*): (1) Dynamic parameter search space construction: the user is allowed to dynamically construct the hyperparameter search space for each model and then generate the objective function based on the interaction with the experimental object, and the next search strategy is based on the historical evaluation of previous experiments to determine the next set of hyperparameters to be validated. The next stage of the search strategy determines the next set of hyperparameters to be validated based on the historical evaluation of previous experiments. (2) Efficient optimization algorithms: including the state-of-the-art hyperparameter sampling algorithm and the pruning algorithm for terminating hopeless trails. Both relational and independent sampling techniques are used to quickly determine the optimal search space. Pruning algorithms are able to end hopeless trails early in the training, also known as automatic early stop. These two types of optimization algorithms are essential for performance optimization in resource-limited situations. (3) Generic architecture and simple setup: It is suitable for many types of experimental purposes at the same time and achieves the requirement of deploying lightweight experiments to large-scale distributed computing through an interactive interface with the easiest setup.

---

**Algorithm 1:** Pseudocode of Optuna Hyperparameter optimization process

---

    **Input:** Dataset $D$, Trail number, Hyperparameter and its value range ;

    **Output:** The maximum accuracy and its corresponding trail ;

1  **procedure** Dataset preprocessing ;

2  Convert symbolic features in the dataset to one-hot code ;

3  Numerically represent the symbolic feature columns in the dataset based on the
    type of attack ;

4  Min-max normalization ;

5  $D_n \leftarrow$ Reduce the dataset features by IPCA ;

6  $D_a \leftarrow$ The training dataset divided from $D_n$ ;

7  $D_b \leftarrow$ The testing dataset divided from $D_n$ ;

8  **return** $D_a, D_b \Rightarrow$ the k-dimensional feature space ;

9  **procedure** Objective function definition ;

10  Define the hyperparameter and its search space ;

11  Input $D_a$ into the DNN-BiLSTM model for training;

12  DL-BiLSTM $\leftarrow$ Dynamic quantization model after training ;

13  Input $D_b$ into the DL-BiLSTM model for testing ;

14  $y_{pred} \leftarrow$ The predicted value of the DL-BiLSTM based on the input ;

15  $y_{test} \leftarrow$ The true value of the output in the testing dataset ;

16  $A_t \leftarrow$ The accuracy score obtained by $y_{pred}$ and $y_{test}$;

17  return $A_t \Rightarrow$ Accuracy of the DL-BiLSTM on the testing dataset ;

18  **procedure** Optimization process creation ;

19  Define the optimization direction to maximize ;

20  Input objective function ;

21  Input trail number ;

22  **return** The optimal value of each hyperparameter under the maximum $A_t$ ;

---

Algorithm 1 describes the process of Optuna hyperparameter optimization. The inputs to the algorithm include the experimental dataset, the number of trials and the hyperparameters to be optimized and their search space. The input raw dataset is first subjected to a series of preprocessing operations to divide the IPCA-dimensioned data into a training dataset and a test dataset (lines 1 to 8 of the algorithm). Then the objective function of optimization is defined. In the optimization process of the detection approach suggested in this study, the input hyperparameters to be optimized include batch_size, learning rate, and weight_decay. It is necessary to set the search space for these hyperparameters in the definition of the objective function to facilitate the subsequent sampling of hyperparameters (line 10). The test accuracy after model training and testing is used as the return value of the objective function to evaluate the detection performance of each trial (lines 11 through 17). Next, the optimization process is created, given whether the direction of the optimization is to find the maximum or minimum value of the return value of the objective function. In this article, we need to find the hyperparameter value that satisfies the maximum value of the test accuracy in the specified search space (lines 18 to

19). Finally, the optimization is started by setting the number of trials to be performed and Optuna will keep approaching the optimal combination of hyperparameters in the given hyperparameter search space through the hyperparameter sampling algorithm based on the performed trials. The trials with the highest detection accuracy and the corresponding hyperparameter values are displayed in the output (lines 20 to 22).

## DATASETS AND PREPROCESSING

In this study, we have chosen three widely recognized public datasets for intrusion detection to serve as benchmark datasets for IoT. We conducted anomaly detection experiments to evaluate the efficacy of our proposed intrusion detection model for IoT intrusion detection. Below, we provide a description of the datasets used, as well as the details of the preprocessing stages.

### Dataset description

Before delving into the data preprocessing methods, we will provide a brief introduction to the three intrusion detection datasets utilized in our proposed classification model.

1. CIC IDS2017 dataset: The network traffic characteristics in the CIC IDS2017 dataset were extracted by the Canadian Institute for Cyber Security through CICFlowMeter software, and the raw network traffic data were captured over a five-day period lasting from Monday morning to Friday afternoon (*Sharafaldin, Lashkari & Ghorbani, 2018*). It is designed for research in network security and intrusion detection techniques, covers a variety of attack scenarios including some common updated attack families such as DoS, DDoS, Brute Force, XSS, SQL Injection, Infiltration, Port scan, and Botnet, and meets the 11 criteria of the dataset evaluation framework. In this article, we adopt the dataset captured and generated during the day of Friday as the IoT intrusion detection dataset. The detailed descriptions of the CIC IDS2017 datasets used are shown in Table 1.

2. N-BaIoT dataset: The N-BaIoT dataset is a benchmark dataset downloaded from the Machine Learning Library at the University of California, Irvine (*Meidan et al., 2018*). This dataset generates malicious IoT traffic by infecting nine Linux-based commercial IoT devices with the two most common IoT botnets, BASHLITE and Mirai. Each IoT device uses a deep autoencoder to extract network behavior snapshots for identifying abnormal IoT behavior and learning normal behavior. In this study, the N-BaIoT dataset is used to simulate a more realistic IoT network attack environment and evaluate the model's ability to detect IoT intrusions as one of the benchmark datasets. The dataset contains approximately 7,062,606 instances with real attributes, covering benign IoT network traffic samples and samples of attacks from IoT botnets. This article uses specified instances of selected IoT devices, including ACK, Scan, SYN, UDP, and UDPP flooding attacks, and a detailed description of the dataset can be found in Table 2.

3. CICIoT2023 dataset: The CICIoT2023 dataset represents a pioneering and realistic collection of IoT attacks, developed by *Neto et al. (2023)*. It employs a diverse set of topologies, incorporating 105 authentic IoT devices to emulate the deployment of IoT

**Table 1  CIC IDS2017 network traffic distribution.**

| Attack category | Training dataset | Testing dataset |
| --- | --- | --- |
| BENIGN | 265326 | 82785 |
| DDos | 81778 | 25651 |
| PortScan | 101709 | 31808 |
| Bot | 1263 | 405 |
| Total | 450076 | 140649 |

**Table 2  N-BaIoT network traffic distribution.**

| Attack category | Training dataset | Testing dataset |
| --- | --- | --- |
| BENIGN | 35625 | 7499 |
| ACK | 73675 | 15221 |
| Scan | 77836 | 16105 |
| SYN | 88639 | 18494 |
| UDP | 171838 | 35585 |
| UDPP | 59327 | 12344 |
| Total | 506940 | 105248 |

products and services within a smart home environment. Thirty-three distinct attacks were executed, recorded, and compiled on this IoT topology. These attacks fall into seven categories: DDoS, DoS, Recon, Web-based, brute force, spoofing, and Mirai. The Recon attack gathers comprehensive information about the target from the IoT topology, whereas the Mirai attack system involves a massive DDoS attack targeting IoT devices. Both categories are typical and novel attack classes in IoT traffic, thus this article focuses on these two main attack classes and their variants for experimentation. For further details about the dataset, please refer to Table 3.

## Data preprocessing

The experimental dataset needs to be preprocessed to comply with the input format of the neural network before being applied to the network intrusion detection model suggested in this article. The data preprocessing process of this intrusion detection system consists of four stages, as described below.

1. Symbolic feature numerization: This process entails converting symbolic features in the dataset (*e.g.*, protocol types, flags, and services) into one-hot codes.
2. Label numerization: The symbolic class feature column in the dataset file is represented as a numerical value according to the attack type. For dichotomous experiments, the character data are represented by the numerical value 0 or 1, the normal class is classified as 0 and the attack class is classified as 1. For multiclassification experiments, the numerical value representing the character label will be increased, taking the CIC IDS2017 dataset as an example, 0 represents the normal class, 1 represents DDos, 2 indicates PortScan, and so on.
3. Data normalization: Since the dimensionality of each feature in the intrusion detection dataset we use varies greatly, using the original dataset directly for detection analysis

**Table 3 CICIoT2023 network traffic distribution.**

| Attack category | Training dataset | Testing dataset |
| --- | --- | --- |
| BENIGN | 138542 | 39837 |
| Recon-PortScan | 11379 | 1952 |
| Recon-OSScan | 14965 | 980 |
| Recon-HostDiscovery | 18448 | 3286 |
| Recon-PingSweep | 335 | 2 |
| Mirai-greeth_flood | 130796 | 29478 |
| Mirai-udpplain | 115633 | 28787 |
| Mirai-greip_flood | 95402 | 27056 |
| Total | 525500 | 131378 |

would result in indicators with higher numerical levels occupying a higher weight in the detection analysis. To effectively train the neural network model, the most important step we must perform in data preprocessing is data normalization, which requires the use of data normalization methods to limit the feature values to a certain range to guarantee the accuracy of data detection. The min-max normalization used in this study normalizes the value of each feature to a range between 0 and 1. The equation for this method is expressed as

$$x_{normalized} = \frac{x - x_{min}}{x_{max} - x_{min}} \tag{21}$$

where $x$ denotes the current value of the data feature and $x_{min}$ and $x_{max}$ denote the minimum and maximum values of the data feature, respectively.

4. Feature dimensionality reduction: The high-dimensional features in the original dataset increase the workload of the classification process of the intrusion detection model. For the lightweight characteristics of the intrusion detection model proposed in this article, it is necessary to choose a suitable feature dimensionality reduction algorithm to retain the low-dimensional, high-quality data input. In this article, the IPCA dimensionality reduction algorithm is used to quickly and efficiently transform high-dimensional features into low-dimensional features through (linear/nonlinear) mapping functions, and the dimensionality reduction principle of this method is described in detail in Section 3.1.

## EXPERIMENTAL RESULTS AND ANALYSIS

Comprehensive experiments are carried out in this part to demonstrate the superiority of the detection performance of the lightweight IoT model proposed in this article in the IoT environment, and all experiments will be conducted on three intrusion detection datasets: CIC IDS2017, N-BaIoT, and CICIoT2023. Before conducting the experiments, we briefly introduce the environment in which the experiments are conducted and the evaluation metrics used.

1. Experimental environment: All evaluations were performed in Python 3.8, using the PyTorch framework, running on a PC with Windows 10 OS, Intel Core i5-7200U CPU @2.50 GHz and 12 GB RAM.

2. Evaluation metrics: This article uses accuracy, precision, recall, and $F_1$-score as evaluation metrics for the IoT intrusion detection model because carrying out intrusion detection in the IoT environment is complicated (*Powers, 2020*). The parameters used for evaluation and the related calculation formulae are shown below.

$$Accuracy = \frac{T_p + T_n}{T_p + T_n + F_p + F_n} \tag{22}$$

$$Precision = \frac{T_p}{T_p + F_p} \tag{23}$$

$$Recall = \frac{T_p}{T_p + F_n} \tag{24}$$

$$F_1 = \frac{2 Precision \cdot Recall}{(Precision + Recall)} = \frac{2 T_p}{2 T_p + F_p + F_n} \tag{25}$$

where $T_p$ is true positive, $T_n$ is true negative, $F_p$ is false positive, and $F_n$ is false negative. Additionally, we took into account evaluation measures like model size, model complexity, and parameter count as a sign of the computational resources of the model.

## Analysis of performance metrics on the CIC IDS2017 dataset
### Comparison of dimensionality reduction algorithms

This experiment compares and analyzes the detection accuracy with the principal component analysis (PCA) and independent component analysis (ICA) dimensionality reduction methods, which are frequently used in intrusion detection, in order to demonstrate the superiority of the IPCA dimensionality reduction algorithm. Figure 6 shows the relationship between the detection accuracy and dimensionality of the three dimensionality reduction algorithms on the CIC IDS2017 dataset. The network traffic of the CIC IDS2017 dataset after data preprocessing is 78 dimensions in total, so the dimensionality of the feature vectors in our comparison experiments of the dimensionality reduction algorithms is selected between 1 and 78, and the increment is taken as 10.

From Fig. 6, it can be seen that the detection accuracy of the two dimensionality reduction algorithms, PCA and ICA, fluctuates greatly in the low-dimensional case, starts to rise steadily after 30 dimensions, and finally stabilizes. The IPCA dimensionality reduction algorithm maintains a high detection accuracy, becomes increasingly accurate with the increasing number of dimensions, and finally remains stable within the critical value. The overall detection accuracy of all three dimensionality reduction algorithms increases with increasing dimensionality and starts to stabilize after reaching a certain number of dimensions, after which the detection accuracy may tend to decrease with increasing dimensionality. The increase in detection accuracy in the early stage is because the increase in feature dimensionality in a certain dimensional range allows the model to learn more effective information, which is beneficial to the detection of data. The decline in the later stage is because the high-dimensional network traffic may bring redundancy and noise, causing the degradation of detection performance.

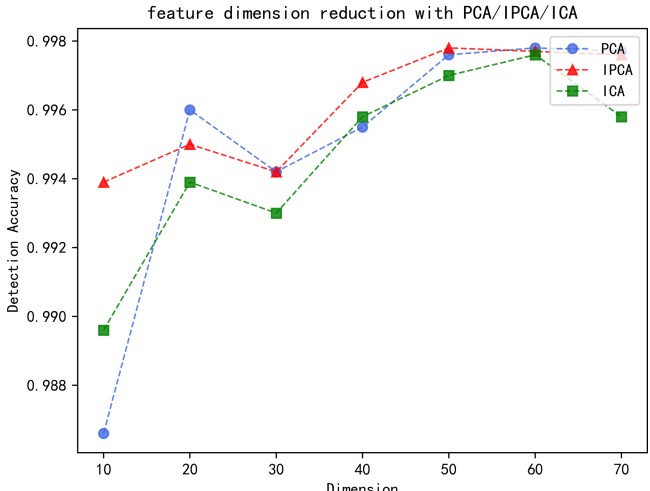

**Figure 6** **Detection accuracy of each dimensionality reduction algorithm.**

This experiment also compares the optimal detection performance of the three dimensionality reduction algorithms on the CIC IDS2017 data, where the IPCA algorithm outperforms the other two algorithms in terms of accuracy, precision, recall and $F_1$-score, and the detailed performance comparison can be seen in Fig. 7. In summary, the IPCA algorithm selected in this article outperforms the current commonly used dimensionality reduction algorithms in the experiment and exhibits excellent performance.

### Performance comparison with other models

To verify the feasibility of the proposed model for IoT intrusion detection, this subsection compares the DL-BiLSTM model with the submodels (the DNN model and the BiLSTM model) fused into it for experiments. Additionally, it compares the proposed model with other commonly used deep learning and machine learning models in terms of accuracy, precision, recall, $F_1$-score, training time, and inference time. The results of the comparison experiments are shown in Table 4. Due to the inability to accurately calculate the training and inference time of machine learning models, "-" is used in Table 4.

As shown in Table 4, the accuracy, precision, recall, and $F_1$-score of the proposed DL-BiLSTM model are the best among all models, with all performance evaluation indexes above 99.5%. This is mainly because the DL-BiLSTM model incorporates the feature extraction capability of both DNN and BiLSTM local algorithms and is able to learn a more comprehensive feature representation. It can be seen that the BiLSTM model itself has excellent performance detection indexes, but the model structure is more complex resulting in a less lightweight model that requires more computational resources. After dynamic quantization, the size of the BiLSTM model is reduced to nearly one-third of the original size, but the model's detection performance degrades as the quantization of the specified cell structure. However, the DNN model structure in DL-BiLSTM can compensate for the shortcomings of the BiLSTM model structure by enabling more adequate feature

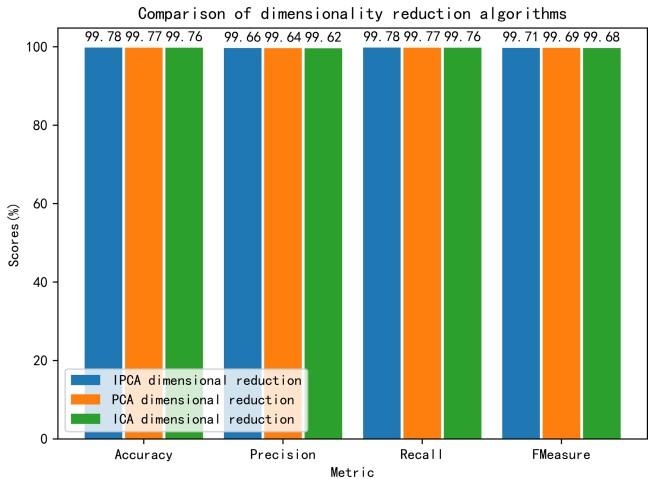

**Figure 7**   **Comparison of the detection performance of each dimensionality reduction algorithm.**

**Table 4   Results of multiclassification comparison of different models on the CIC IDS2017 dataset.**

| Model | Accuracy | Precision | Recall | $F_1$ | Training time(s) | Inference time(s) |
|---|---|---|---|---|---|---|
| DNN | 0.9954 | 0.9941 | 0.9954 | 0.9947 | 536.0 | 6.5 |
| BiLSTM | 0.9958 | 0.9946 | 0.9958 | 0.9951 | 688.4 | 7.2 |
| RNN | 0.9952 | 0.9939 | 0.9952 | 0.9945 | 575.3 | 7.5 |
| CNN | 0.9963 | 0.9949 | 0.9963 | 0.9956 | 1194.6 | 7.2 |
| SVM | 0.9800 | 0.9780 | 0.9800 | 0.9787 | – | – |
| MLP | 0.9956 | 0.9954 | 0.9956 | 0.9955 | – | – |
| DL-BiLSTM | 0.9967 | 0.9954 | 0.9967 | 0.9959 | 746.2 | 6.6 |

extraction so that it can achieve better classification detection capability than both of the original local algorithms with less computational resources.

The DL-BiLSTM model has the shortest inference time relative to the other models, while the training time is long relative to DNN, RNN and BiLSTM. This is due to the fact that the basic model architecture of DL-BiLSTM is an integrated model that fuses DNN and BiLSTM together, and the training phase uses a model that has not yet been quantized. Compared to most single models, there is not much computational processing advantage during the training phase of the model. However, this advantage becomes apparent during the inference phase when quantization processing is utilized.

Figure 8 shows the prediction results of the DL-BiLSTM model and other comparative models on each sample category in the CIC IDS2017 dataset, presented in the form of a confusion matrix. The confusion matrix intuitively represents the performance of the model. The darker the color of a square, the more samples are correctly predicted for that corresponding category, indicating higher classification accuracy. Importantly, we can visually obtain the false positive rate of model detection from the confusion matrix, which refers to the number of negative samples incorrectly identified as benign samples. From Fig. 8, it can be seen that compared to BiLSTM and RNN models, DL-BiLSTM model

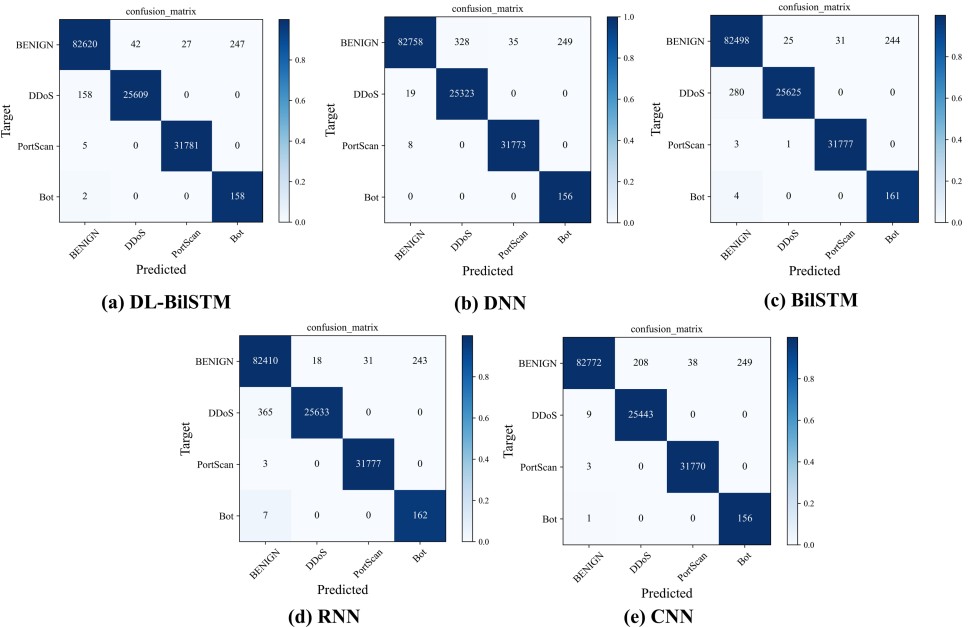

**Figure 8  Confusion matrix for each model on the CIC IDS2017 dataset.**

**Table 5  Multiclassification comparison results of different models on the N-BaIoT dataset.**

| Model | Accuracy | Precision | Recall | $F_1$ | Training time(s) | Inference time(s) |
|-------|----------|-----------|--------|-------|------------------|-------------------|
| DNN-LSTM | 0.9924 | 0.9928 | 0.9924 | 0.9925 | 799.2 | 6.4 |
| PCA+LSTM | 0.9847 | 0.9851 | 0.9847 | 0.9847 | 1081.4 | 6.2 |
| BiLSTM | 0.9921 | 0.9924 | 0.9921 | 0.9922 | 1039.1 | 6.1 |
| DL-BiLSTM | 0.9998 | 0.9998 | 0.9998 | 0.9998 | 1154.7 | 6.0 |

exhibits a lower false positive rate and is able to effectively identify most attack samples. Although it may not perform as well as DNN and CNN models in terms of false positives, DL-BiLSTM model has an advantage in terms of false negatives. Therefore, our proposed DL-BiLSTM model achieves excellent detection accuracy and accurately predicts most experimental sample categories.

## Analysis of performance metrics on the N-BaIoT dataset

Comparative experiments using the intrusion detection method proposed in this article with the intrusion detection method in the literature (*Qureshi et al., 2021*; *Zhongshi, Yan & Yudong, 2019*; *Esmaeili et al., 2022*) on the N-BaIoT dataset with multiple classifications are shown in Table 5. From the detection results in the table, we can see that the performance detection index of each method proposed in this article is as high as 99.98% in the N-BaIoT dataset, and the detection performance is significantly better than several others. In particular, compared with the LSTM model using the PCA dimensionality reduction method, the accuracy rate is improved by 1.51%.

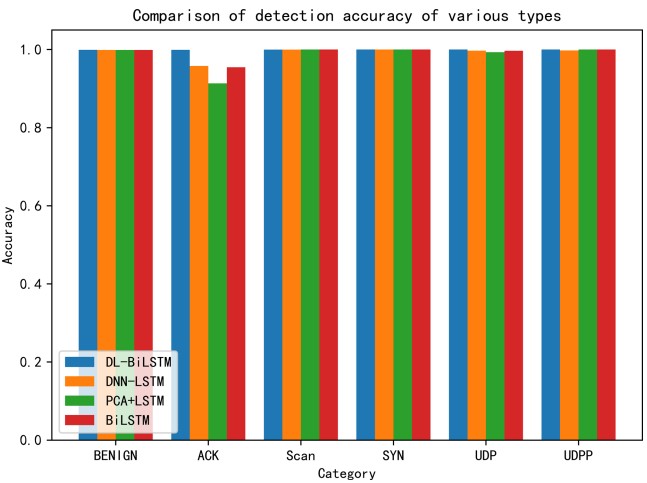

**Figure 9  Comparison of detection accuracy by category.**

The training time of DL-BiLSTM is longer compared to that of the other methods, while the testing time does not exhibit a significant advantage over these methods. This difference in training time could be attributed to the fact that the comparison methods utilize either a single LSTM or BiLSTM model, while our proposed method employs an integrated DNN and BiLSTM model. The integration can lead to increased computational complexity during training. However, during the testing phase, the proposed method maintains its efficiency due to the quantization process.

For a more thorough analysis of the detection performance of the suggested method on the N-BaIoT dataset, the detection accuracy of the suggested strategy is contrasted against the other methodologies for each category of the dataset in Table 5. The comparison findings are displayed in Fig. 9. The accuracy of the proposed method in the scan category is 0.01% lower than that of the last two methods, and the accuracy of the proposed method in other categories reaches the highest level compared with the other three methods. In particular, the detection rates of the ACK and UDP categories are significantly higher than those of the other methods.

Figure 10 further shows the specifics of the multiclassification detection results of the above models on the N-BaIoT dataset in the form of a confusion matrix. As shown in Fig. 10, DL-BiLSTM has the best classification results, especially in the classification detection of three sample classes, SYN, UDP, and UDPP, where DL-BiLSTM achieves 100% detection accuracy, and almost all other classes of samples can be successfully predicted. The other three comparison models perform extremely poorly in detecting ACK as an attack class, and a large proportion of ACK samples are identified and predicted as UDP. In addition, the LSTM model based on the PCA dimensionality reduction algorithm identifies a proportion of UDP samples as ACK. All models, except for the DNN-LSTM model, performed well in terms of false positives, with a false positive rate of 0.

The proposed approach in this article has great results and outperforms other methods according to both the comparative experimental results of each performance index of

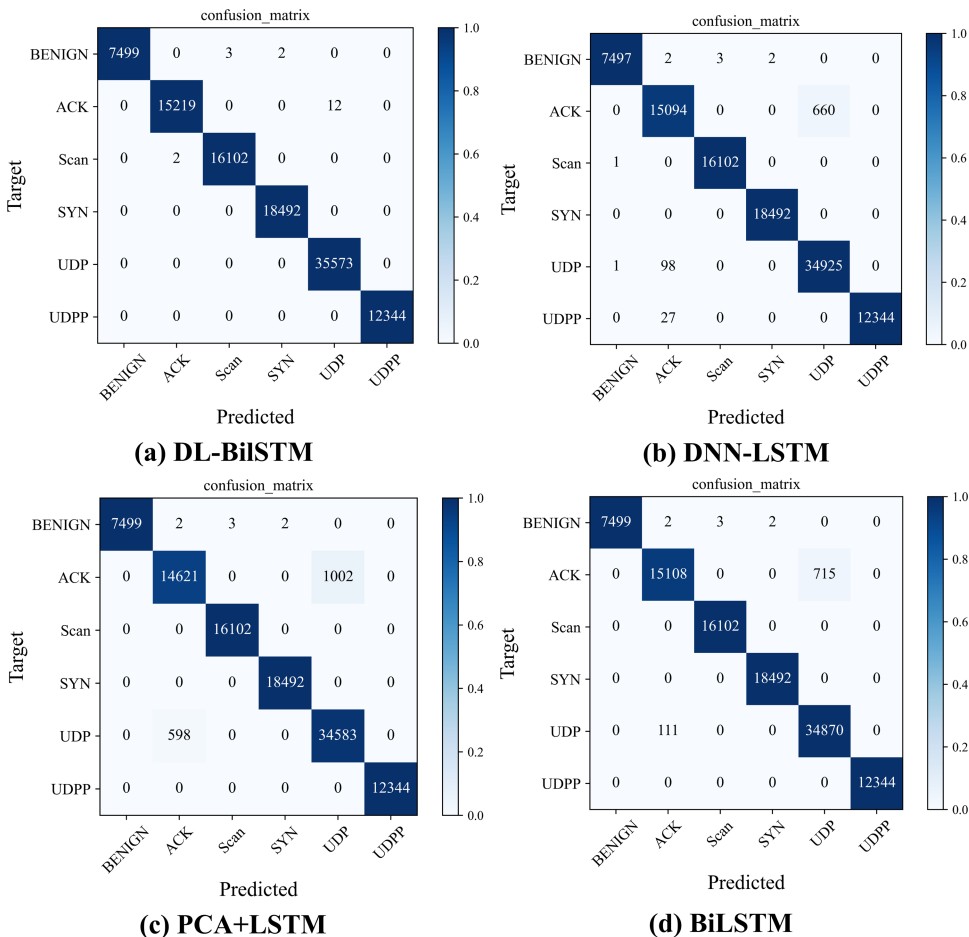

**Figure 10   Confusion matrix of each model on the N-BaIoT dataset.**

multicategorization and the comparative results of the detection accuracy of each category. This is because both the LSTM model based on PCA dimension reduction (PCA+LSTM) and the intrusion detection method integrating DNN and LSTM (DNN-LSTM) can only extract one-way feature information and cannot effectively mine data features. While the BiLSTM is able to extract bidirectional features, it does not sufficiently extract nonlinear features and cannot achieve higher feature extraction performance compared with the DL-BiLSTM.

## Analysis of performance metrics on the CICIoT2023 dataset

To verify the effectiveness of the DL-BiLSTM model in detecting new IoT network attacks, this article evaluates the method using the latest IoT dataset CICIoT2023 and compares it with other classical deep learning models. The comparison results are shown in Table 6. The experimental results indicate that the proposed DL-BiLSTM model outperforms other models in detecting recent network data attacks across all performance metrics. Additionally, this model has the shortest training and inference time, especially compared

**Table 6  Multiclassification results of the CICIoT2023 dataset.**

| Model | Acurracy | Precision | Recall | $F_1$ | Training time(s) | Inference time(s) |
|---|---|---|---|---|---|---|
| CNN | 0.9221 | 0.9149 | 0.9221 | 0.9126 | 1515.4 | 7.2 |
| RNN | 0.9273 | 0.9124 | 0.9273 | 0.9150 | 717.8 | 8.5 |
| LSTM | 0.9275 | 0.9132 | 0.9275 | 0.9152 | 764.8 | 6.6 |
| BiLSTM | 0.9305 | 0.9133 | 0.9305 | 0.9173 | 792.6 | 8.0 |
| DL-BiLSTM | 0.9313 | 0.9180 | 0.9313 | 0.9194 | 708.4 | 6.4 |

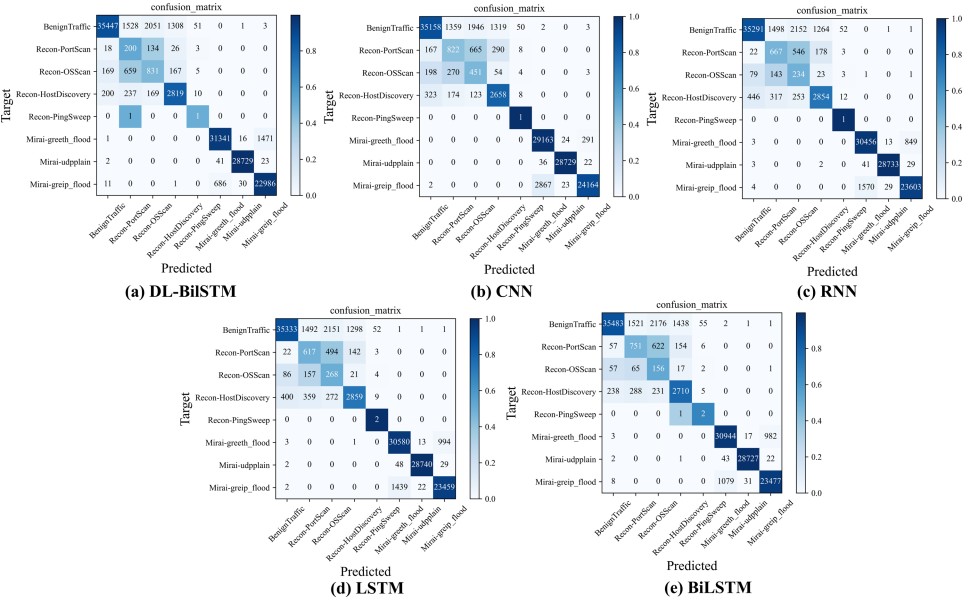

**Figure 11  Confusion matrix of each model on the N-BaIoT dataset.**

to CNN models, where it has a significant advantage in training time, being only half of CNN's.

Figure 11 shows the confusion matrix results of the DL-BiLSTM model and other comparative models for multi-class detection on the CICIoT2023 dataset. From the figure, it can be observed that all models tend to have confusion in detecting the Recon-PortScan and Recon-OSScan categories, as these two attack categories have similar feature information, making it difficult for the models to differentiate between them. The same issue also occurs in detecting Mirai-greip flood and Mirai-greeth flood, but compared to other models, DL-BiLSTM performs better in detecting these two attack categories. In terms of false positives, DL-BiLSTM outperforms CNN, RNN, and LSTM models and has a false positive rate comparable to BiLSTM.

## Confusion matrix for each model on the CICIoT2023 dataset

We compared the DL-BiLSTM model with recent studies (*Shieh et al., 2021*; *Wang et al., 2021*; *Bhardwaj, Mangat & Vig, 2020*; *Qazi, Almorjan & Zia, 2022*; *Qureshi et al., 2021*; *Alharbi et al., 2021*; *Attique, Hao & Ping, 2022*; *Om Kumar et al., 2022*) on two datasets,

**Table 7  Multiclassification results of the CIC IDS2017 dataset.**

| Model | Accuracy | Precision | Recall | $F_1$ |
|---|---|---|---|---|
| BI-LSTM-GMM | 0.9820 | 0.9790 | 0.9980 | 0.9880 |
| Improved BYOL | 0.9670 | 0.9500 | 0.9596 | 0.9548 |
| AE+DNN | 0.9892 | 0.9745 | 0.9897 | 0.9835 |
| 1D-CNN | 0.9896 | 0.9870 | 0.9920 | 0.9894 |
| DL-BiLSTM | 0.9967 | 0.9954 | 0.9967 | 0.9959 |

**Table 8  Multiclassification results for the N-BaIoT dataset.**

| Model | Accuracy | Precision | Recall | $F_1$ |
|---|---|---|---|---|
| DNN-LSTM | 0.9996 | 0.9977 | 0.9966 | 0.9966 |
| LGBA-NN | – | 0.8523 | 0.9000 | 0.8664 |
| Cu-DNNGRU | 0.9939 | 0.9909 | 0.9889 | 0.9921 |
| RKCNN-MMBO | 0.9996 | 0.9985 | 0.9991 | 0.9988 |
| DL-BiLSTM | 0.9998 | 0.9998 | 0.9998 | 0.9998 |

CIC IDS2017 and N-BaIoT, in order to further illustrate the superiority of the detection performance of the model provided in this article. The experimental results can be seen in Tables 7 and 8, and the performance test results not given in the paper are indicated with "-" in the table. The DL-BiLSTM model suggested in this research performs detection better than existing state-of-the-art methods, as shown in Tables 7 and 8. In Table 7, the recall rate of the DL-BiLSTM model is only 0.13% lower than that of BI-LSTM-GMM because BI-LSTM-GMM is a detection model based on BiLSTM and the Gaussian mixture model (GMM), which has more parameters and a more intricate model framework. However, the DL-BiLSTM model can still outperform BI-LSTM-GMM in accuracy, precision, and $F_1$-score in the case of a smaller number of parameters and lower model complexity. In particular, the DL-BiLSTM model improved 4.54% over the improved BYOL model and 1.64%, 2.09%, and 0.84% over the BI-LSTM-GMM, AE+DNN, and 1D-CNN models, respectively, in terms of precision. For the $F_1$-score, the DL-BiLSTM model obtains 99.59%, which is 0.79%, 4.11%, 1.24% and 0.65% better than the BI-LSTM-GMM, improved BYOL, AE+DNN and 1D-CNN models, respectively. This result shows that the DL-BiLSTM model can guarantee lightweight and effective network intrusion detection. In Table 8, the DL-BiLSTM model outperforms the other methods in the table in all performance metrics. In terms of precision, the DL-BiLSTM model is 0.21%, 14.75%, 0.89% and 0.13% better than the DNN-LSTM, LGBA-NN, Cu-DNNGRU and RKCNN-MMBO models, respectively, and the improvement in $F_1$-score is 0.32%, 13.34%, 0.77% and 0.1%, respectively, further proving the effectiveness of the proposed method in this article for IoT intrusion detection.

## Lightweight performance analysis of DL-BiLSTM

The suggested method in this study emphasizes on the lightweight of the model in addition to the pursuit of outstanding classification performance because it takes into account the practical element of the restricted computational resources of IoT devices. In this subsection, we experimentally compare the model size and model complexity of different

**Table 9  Comparison of model size and computational cost on the CIC IDS2017 dataset.**

| Model | Parameters | FLOPs | Model size |
|---|---|---|---|
| DNN | 10692 | 4354048 | 45.7KB |
| BiLSTM | 16132 | 7143936 | 68.8KB |
| RNN | 16204 | 7150800 | 68.3KB |
| CNN | 18372 | 9319936 | 83.5KB |
| DL-BiLSTM | 1654 | 610800 | 28.3KB |

**Table 10  Comparison of model size and computational cost on the N-BaIoT dataset.**

| Model | Parameters | FLOPs | Model size |
|---|---|---|---|
| DNN-LSTM | 11206 | 6541136 | 48.8KB |
| PCA+LSTM | 23430 | 13755136 | 98.3KB |
| BiLSTM | 12438 | 7326368 | 54.5KB |
| DL-BiLSTM | 1742 | 785568 | 33KB |

**Table 11  Comparison of model size and computational cost on the CICIoT2023 dataset.**

| Model | Parameters | FLOPs | Model size |
|---|---|---|---|
| CNN | 42952 | 31322240 | 184.1KB |
| RNN | 15608 | 6802400 | 65.8KB |
| LSTM | 21128 | 9209280 | 88.4KB |
| BiLSTM | 13928 | 6070080 | 59.9KB |
| DL-BiLSTM | 1988 | 628800 | 36.5KB |

deep learning methods proposed earlier. Among them, the evaluation metric of floating point operations (FLOPs) is introduced, which indicates the computational volume of the model and can be used to measure the complexity of the model. As CIC IDS2017, N-BaIoT, and CICIoT2023 have different numbers of data features, this experiment is being conducted to assess the model's lightweight effectiveness on the three datasets separately. Specific experimental results can be seen in Table 9, Table 10, and Table 11.

Table 9, Table 10, and Table 11 show that the DL-BiLSTM model proposed in this article has the smallest model size and the lowest model complexity on the above three datasets. On the CIC IDS2017 dataset, the proposed method reduces the complexity by 85.97% compared to the DNN model with the simplest model structure, and the model size is only 34% of that of CNN. In particular, on the N-BaIoT dataset, the model complexity and model size of the proposed model are reduced by 94.29% and 66.43%, respectively, compared with the LSTM model based on the PCA algorithm. On the CICIoT2023 dataset, the model complexity and model size of DL-BiLSTM are only 20% of CNN. This demonstrates the outstanding lightweight of the suggested method in this article for IoT intrusion detection. The proposed method uses little computational resources, and there are two key reasons for this: first, the complexity of the model is decreased by mapping the high-dimensional data space to the low-dimensional space during the data preprocessing step using the IPCA dimensionality reduction algorithm; second, the dynamic quantization of the specified cell

structure of the model makes the proposed method have a small model size, which can be deployed on IoT devices with limited computational resources.

## CONCLUSION

In order to address the demands of large-scale, intricate IoT intrusion detection network attacks as well as lightweight and real-time detection, this article provides a hybrid intrusion detection model that combines DNN and BiLSTM. On the basis of this model, a lightweight IoT intrusion recognition system is also created. The suggested approach employs an IPCA algorithm to carry out feature dimensionality reduction during the data preprocessing stage, extracts nonlinear and bidirectional long-range features of network data using DNN-BiLSTM, and dynamically quantifies the specified unit structure of the model at the completion of training to obtain a DL-BiLSTM lightweight intrusion detection model. Experiments are carried out using the benchmark datasets CIC IDS2017, N-BaIoT, and CICIoT2023 to simulate the industrial and network environments of the IoT in order to evaluate the effectiveness of the suggested method. The experimental findings demonstrate that, in terms of detection performance, the suggested approach exceeds existing detection methods on both datasets. In addition, it has a smaller model size and lower model complexity than existing detection methods.

Although the approach suggested in this study has good overall detection performance and is lightweight, there are some limitations to consider. First, the posttraining dynamic quantization method may not be able to fully restore the best detection state of the original model at the time of training, and it also increases the intrusion detection steps. Second, the experimental design of the intrusion detection dataset cannot fully replicate the complex and changing network environment of the IoT. Therefore, the generalizability of this study has some limitations. Future research will concentrate on investigating advanced quantization processing techniques to incorporate model quantization into the training process, improving the overall effectiveness of the intrusion detection system, and better restoring the detection performance of the lightweight model. To improve the generality of the intrusion detection model, we will continue to broaden the dataset in upcoming experimental research to incorporate a wider variety of IoT devices and real network scenarios.

### Funding

This work is supported by the Natural Science Foundation of China (62062037, 61562037, 72261018), and the Natural Science Foundation of Jiangxi Province (20212BAB202014, 20171BAB202026). The funders had no role in study design, data collection and analysis, decision to publish, or preparation of the manuscript.

### Grant Disclosures

The following grant information was disclosed by the authors:

The Natural Science Foundation of China: 62062037, 61562037, 72261018.
The Natural Science Foundation of Jiangxi Province: 20212BAB202014, 20171BAB202026.

## Competing Interests

The authors declare that there are no competing interests.

## Author Contributions

- Zhendong Wang conceived and designed the experiments, performed the experiments, analyzed the data, prepared figures and/or tables, authored or reviewed drafts of the article, and approved the final draft.
- Hui Chen conceived and designed the experiments, performed the experiments, analyzed the data, performed the computation work, prepared figures and/or tables, authored or reviewed drafts of the article, and approved the final draft.
- Shuxin Yang conceived and designed the experiments, authored or reviewed drafts of the article, and approved the final draft.
- Xiao Luo performed the experiments, prepared figures and/or tables, and approved the final draft.
- Dahai Li analyzed the data, authored or reviewed drafts of the article, and approved the final draft.
- Junling Wang conceived and designed the experiments, analyzed the data, authored or reviewed drafts of the article, and approved the final draft.

## Data Deposition

The source code is available in the Supplemental File.

The intrusion detection datasets CIC IDS2017 and N-BaIoT that were used in this study are available at, respectively:

- http://205.174.165.80/CICDataset/CIC-IDS-2017/Dataset/
- https://archive.ics.uci.edu/ml/datasets/detection_of_IoT_botnet_attacks_N_BaIoT
- http://205.174.165.80/IOTDataset/CIC_IOT_Dataset2023/CICIOT/

## Supplemental Information

Supplemental information for this article can be found online at http://dx.doi.org/10.7717/peerj-cs.1569#supplemental-information.

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
