# Peer review of "A lightweight intrusion detection method for IoT based on deep learning and dynamic quantization"

_PeerJ Computer Science, doi:10.7717/peerj-cs.1569_

## Round 0.1 · original submission · Major Revisions

The authors should prepare a revision while addressing all the reviewer comments.

Both reviewers have requested that you cite specific references. You may add them if you believe they are especially relevant. However, I do not expect you to include these citations, and if you do not include them, this will not influence my decision.

·

Basic reporting

1. Your introduction needs more explanation. I suggest that you improve the description in lines 39 - 40 to provide more justification for your statement (intrusion detection systems are composed of three primary components: data collection, conversion to select features, and decision engines).


2. The literature review section includes a good amount of recent studies but is limited to older datasets. I suggest that you explore further that has newer datasets in the studies.

3. You introduced Evaluation metrics in line 488. I suggest that you cite appropriate study(s). For example, "Evaluation: From Precision, Recall and F-Measure to ROC, Informedness, Markedness and Correlation" by David Power.

Experimental design

1. Y. Meidan et al. (IEEE Pervasive Computing, vol. 17, no. 3, pp. 12-22, Jul.-Sep. 2018, doi: 10.1109/MPRV.2018.03367731) have proposed the N-BaIoT and proposed Deep Autoencoders with good outcomes, but you have not shown the appropriate for this situation in your discussion in line 437 to 448. You explain only about the dataset. Please explain why you used this dataset.

2. Your study has been validated using CICIDS2017 and N-BaIoT, where both datasets were released back in 2018. I suggest that you add newer datasets such as EDGE_IIOT, CIC IoT Dataset 2022, CIC IoT Attack Dataset 2023, etc, into your validation.

3. I appreciate the authors for clearly explaining the methodology section and/or the algorithm.

Validity of the findings

1. You have shown the performance results of the proposed method in Tables 3 and 4 using two datasets. Although your results are compelling, the data analysis should be improved. I suggest that adding the training time and inference time will be more fruitful to proof the validity and to compare your findings with other recent studies.

2. I suggest that the conclusion section include future aspect from this study findings, this will be helpful for the readers.

Additional comments

I suggest that more recent studies on related method be included. Here are some studies, but not limited to:

a) https://doi.org/10.1145/3590003.3590018
b) https://doi.org/10.14569/IJACSA.2023.0140349
c) https://doi.org/10.1016/j.teler.2023.100053
d) https://doi.org/10.3390/s22218417
e) https://doi.org/10.1016/j.jpdc.2023.05.001

Reviewer 2 ·

Basic reporting

The overall structure of the paper is well organized. However, certain significant revisions need to be made prior to its acceptance.
1. The necessity of an intrusion detection system is not adequately explained, particularly in the abstract. A discussion on the need for the proposed system should be included.
2. The introduction is effectively crafted.
3. To enhance this section, the related work should be more comprehensive. It would be beneficial to discuss the following papers:
• DOI: https://doi.org/10.1002/ett.3813).
https://link.springer.com/article/10.1007/s10586-021-03376-3).
• "(DOI: 10.1109/MWC.012.2200373).
4. Several paragraphs within the paper are excessively lengthy, making it difficult to understand the content.
5. The discussion about specific extracted features is missing and should be clearly mentioned.
6. The complexity of the proposed algorithm should be explicitly addressed.
7. It would be beneficial to include a confusion matrix in the dataset description, illustrating the false positive extractions.

Experimental design

Minor revisions needed in the dataset description and extraction phases

Validity of the findings

The results are valid, I have no concerns

Additional comments

Nil

---

## Round 0.2 · accepted · Accept

The authors have addressed the reviewer comments and they have accepted the article.

·

Basic reporting

No Comment.

Experimental design

No comment.

Validity of the findings

The manuscript contains many similar approaches to the following paper:

https://doi.org/10.1016/j.teler.2023.100053

I suggest the authors cite the work where it is applicable, but not compulsorily.

Additional comments

The authors have improved the manuscript, addressing all the concerns raised earlier. The paper is looking better.

Reviewer 2 ·

Basic reporting

the authors have revised the article according to the previous concerns. i have no further concerns

Experimental design

no comment

Validity of the findings

no comment

Additional comments

no comments